# Rethinking Optimization with Differentiable Simulation from a Global Perspective

**Rika Antonova**[1*]    **Jingyun Yang**[1*]    **Krishna Murthy Jatavallabhula**[2]    **Jeannette Bohg**[1]

Stanford University[1]    Massachusetts Institute of Technology[2]

**Abstract:** Differentiable simulation is a promising toolkit for fast gradient-based policy optimization and system identification. However, existing approaches to differentiable simulation have largely tackled scenarios where obtaining smooth gradients has been relatively easy, such as systems with mostly smooth dynamics. In this work, we study the challenges that differentiable simulation presents when it is not feasible to expect that a single descent reaches a global optimum, which is often a problem in contact-rich scenarios. We analyze the optimization landscapes of diverse scenarios that contain both rigid bodies and deformable objects. In dynamic environments with highly deformable objects and fluids, differentiable simulators produce rugged landscapes with nonetheless useful gradients in some parts of the space. We propose a method that combines Bayesian optimization with semi-local 'leaps' to obtain a global search method that can use gradients effectively, while also maintaining robust performance in regions with noisy gradients. We show that our approach outperforms several gradient-based and gradient-free baselines on an extensive set of experiments in simulation, and also validate the method using experiments with a real robot and deformables.

**Keywords:** Differentiable simulation, Global optimization, Deformable objects

## 1 Introduction

Physics simulation is indispensable for robot learning: it has been widely used to generate synthetic training data, explore learning of complex sensorimotor policies, and help anticipate the performance of various learning methods before deploying on real robots. A growing number of works attempt to *invert* physics engines: given simulation outputs (e.g. trajectories), infer the input parameters (e.g. physical properties, robot controls) that best explain the outputs. Differentiable physics engines aim to offer a direct way to invert simulation by enabling gradient-based policy optimization and system identification, e.g. [1, 2, 3, 4, 5]. However, to highlight the most promising aspects, works in this space have focused on scenarios where obtaining smooth gradients is relatively easy.

With more research efforts developing and using differentiable physics engines, it is important to analyze the limits of these systems. Surprisingly, only a few works investigate the *differentiability* of these simulators. Several fundamental limitations of differentiable simulation with rigid contacts in low-dimensional systems are highlighted by [6]. We aim to understand whether such limitations are also prevalent in simulations with deformables. For this, we create several new challenging environments, and also use environments from prior work. We analyze scenarios with flexible deformable objects (cloth), plastics (clay), and fluids. We show that in many cases obtaining well-behaved gradients is feasible, but challenge the assumption that differentiable simulators provide easy landscapes in sufficiently interesting scenarios. Our visualizations of optimization landscapes uncover local optima and plateaus, showing the need for extending gradient-based methods with global search.[†]

We propose a method that combines global search using Bayesian optimization with a semi-local search. Our semi-local strategy allows to make progress on parts of the landscape that are intractable for gradient descent alone. In our experiments we show visual, quantitative and qualitative analysis of the optimization problems arising from simulations with deformables in a variety of scenarios.

---

[*]Rika and Jingyun contributed equally. Contact: {rika.antonova, jingyuny}@stanford.edu

[†]Videos are available at https://tinyurl.com/globdiff

6th Conference on Robot Learning (CoRL 2022), Auckland, New Zealand.

We also validate our proposed approach on a real robot interacting with cloth, where our aim is to identify the properties of the cloth that make the motion of the simulated cloth match the real one.

To summarize, our contributions are: 1) environments with a wide variety of objects and interactions; 2) analysis of landscapes and gradients; 3) a method for global optimization on rugged landscapes.

## 2 Background

**Differentiable Simulation:** The earliest differentiable simulators [7, 8] focused solely on rigid-body dynamics, often operating only over a small number of predefined object primitives. Subsequent methods generalized to arbitrary rigid body shapes [9, 10], added articulations [11, 12, 13] and multiphysics [1]; improved scalability [14], speed [15, 16], and contact modeling [17, 18, 19]. Differentiable simulation has also been explored in the context of deformable objects [20], cloth [21], fluids [22], robotic cutting [23] and other scientific and engineering phenomena [24, 25, 26]. While several approaches sought to produce physically accurate forward simulations and gradients, they did not rigorously study the loss landscapes. One of the main challenges for optimization is that loss landscapes for contact-rich scenarios contain many discontinuities and local optima. This makes the success of gradient descent highly dependent on the starting point (i.e. an initial guess). This is evident even in the most basic case of a single bouncing ball as shown in [16]. In this work, we also verify that the issue is present in the more recent frameworks, such as Nimble [17] and Warp [27].

**Global Search Methods:** Global optimization tackles the problem of finding the global optimum in a given search space. In our case, this could constitute finding the optimal parameters for a controller (e.g. target position, velocity, force, torque) or physical parameters of a simulator to make the behavior of simulated objects match reality. Global optimization includes several broad families of methods: 1) space covering to systematically visit all parts of the space, e.g. [28, 29]; 2) cluster analysis to identify promising areas, e.g. [30, 31]; 3) evolutionary methods that start from a broad set of candidates and evolve them towards exploring the promising regions, e.g. [32, 33]; 4) Bayesian optimization (BO) that can keep track of a global model of the cost function and its uncertainty on the whole search space [34, 35]. Space covering and clustering methods face significant challenges when trying to scale to high dimensions. The survey in [36] gives further details. The recent focus in global optimization has been on evolutionary methods and BO, which can be successful on search spaces with thousands of dimensions. Random search is another method, and one of the few that can guarantee global optimality; it is often not data efficient, but can be a surprisingly robust baseline. Hence, in this work we compare methods based on random search, evolutionary approaches and BO.

**CMA-ES:** One of the most versatile evolutionary methods is Covariance Matrix Adaptation - Evolution Strategies (CMA-ES) [32]. It samples a randomized 'generation' of points at each iteration from a multivariate Gaussian distribution. To evolve this distribution, it computes the new mean from a subset of points with the lowest cost from the previous generation. It also uses these best-performing points to update the covariance matrix. The next generation of points is then sampled from the distribution with the updated mean and covariance. CMA-ES succeeds in a wide variety of applications [37], and has competitive performance on global optimization benchmarks [38]. This method does not make any restrictive assumptions about the search space and can be used 'as-is' i.e. without tuning. However, this method is technically not fully global — while it can overcome shallow local optima, it can get stuck in deeper local optima [39].

**Bayesian optimization (BO):** BO views the problem of global search as seeking a point $\boldsymbol{x}^*$ to minimize a cost function $f(\boldsymbol{x})$, i.e. find $\boldsymbol{x}^*$ such that $f(\boldsymbol{x}^*) = \min_{\boldsymbol{x}} f(\boldsymbol{x})$. BO assumes that evaluation of $f$ is expensive, so tries to only use a small number of $f(\boldsymbol{x})$ evaluations (trials). At each trial, BO optimizes an auxiliary acquisition function to select the next promising $\boldsymbol{x}$ to evaluate. $f$ is frequently modeled with a Gaussian process (GP): $f(\boldsymbol{x}) \sim \mathcal{GP}(m(\boldsymbol{x}), k(\boldsymbol{x}_i, \boldsymbol{x}_j))$. Modeling $f$ with a GP allows to compute the posterior mean $\bar{f}(\boldsymbol{x})$ and uncertainty (variance) $\mathbb{V}[f(\boldsymbol{x})]$ for each candidate point $\boldsymbol{x}$. Hence, the acquisition function can select points to balance a high mean (exploitation) with high uncertainty (exploration). The kernel function $k(\cdot, \cdot)$ encodes similarity between inputs. If $k(\boldsymbol{x}_i, \boldsymbol{x}_j)$ is large for inputs $\boldsymbol{x}_i, \boldsymbol{x}_j$, then $f(\boldsymbol{x}_i)$ strongly influences $f(\boldsymbol{x}_j)$. One of the most widely used kernel functions is the Squared Exponential (SE): $k_{SE}(\boldsymbol{r} \equiv |\boldsymbol{x}_i - \boldsymbol{x}_j|) = \sigma_k^2 \exp\left(-\frac{1}{2}\boldsymbol{r}^T \operatorname{diag}(\boldsymbol{\ell})^{-2}\boldsymbol{r}\right)$. Here, $\sigma_k^2$, $\boldsymbol{\ell}$ are signal variance and a vector of length scales, they are hyperparameters optimized automatically by maximizing the marginal data likelihood. See [35] for further details.

**Problem formulation for policy optimization and real-to-sim:** Global search can optimize parametric policies, with $\boldsymbol{x}$ as a shorthand for a policy $\pi_{\boldsymbol{x}}$ parameterized by a vector of parameters $\boldsymbol{x} \in \mathbb{R}^d$ and $f(\boldsymbol{x})$ as a shorthand for cost (or loss). For reactive policies $\pi_{\boldsymbol{x}}(\boldsymbol{s})$ and state $\boldsymbol{s} \in \mathbb{R}^m$, the aim is to solve : $\min_{\boldsymbol{x}} \mathbb{E}_{\boldsymbol{s_0}, \pi_{\boldsymbol{x}}(\boldsymbol{s}_0),...,\pi_{\boldsymbol{x}}(\boldsymbol{s}_{T-1}), \boldsymbol{s}_T} \left[ \sum_{t \leq T} cost(\boldsymbol{s}_t) \right]$. The expectation here can be both over the system dynamics $\boldsymbol{s}_{t+1} = dyn(\boldsymbol{s}_t, \boldsymbol{a}_t)$ and policy actions $\boldsymbol{a} = \pi_{\boldsymbol{x}}(\boldsymbol{s})$. For example, in locomotion, $\pi_{\boldsymbol{x}}$ could be a reactive stepping controller [40, 41] with parameters $\boldsymbol{x}$; cost (or loss) could be energy used minus distance walked. For open-loop controllers used in this work, $\boldsymbol{x}$ directly encodes control waypoints (e.g. positions, velocities, or forces). For *real-to-sim*, the goal is to find simulator parameters $\boldsymbol{x}$ such that behavior of objects in simulation matches reality. Formally: $\min_{\boldsymbol{x}} \mathbb{E} \left[ \sum_{t \leq T} (dyn^{sim}(\boldsymbol{s}_t^{sim}, \boldsymbol{a}_t^{sim}; \boldsymbol{x}) - s_{t+1}^{real})^2 \right]$. For manipulation with deformables: the important part of $\boldsymbol{s}$ is the state of objects; the expectation can be taken over a single trajectory, since simulation and reality quickly mismatch unless the simulation parameters are inferred correctly.

## 3 BO-Leap: A Method for Global Search on Rugged Landscapes

We propose an approach for global search that can benefit from gradient-based descents and employs a semi-local strategy to make progress on rough optimization landscapes. To explore the loss landscape globally, we use Bayesian optimization (BO). BO models uncertainty over the loss and reduces uncertainty globally by exploring unseen regions. It also ensures to return to low-loss regions to further improve within the promising areas of the search space. BO uses an acquisition function to compute the most promising candidate to evaluate next. We treat each candidate as a starting point for a semi-local search. As we show in our experiments, the straightforward strategy of using gradient descent from each of these starting points does not ensure strong performance in scenarios with noisy gradients. Hence, we propose a hybrid descent strategy that combines gradient-free search with gradient-based descents. For this, we collect a small population of local samples and compute a sampling distribution based on CMA-ES. Instead of directly using the resulting mean and covariance to sample the next population (as CMA-ES would), we use gradient descent to evolve the distribution mean, then use this updated mean when sampling the next population.

We outline our method, BO with semi-local leaps (BO-Leap), in Algorithm 1, and provide full pseudocode in the supplementary. We also visualize the algorithm in Figure 1. We start by initializing a global model for the loss with a Gaussian process (GP) prior. Using the BO acquisition function, we sample a vector $\boldsymbol{x}_1$ of simulation or control parameters to evaluate. We then run a semi-local search from this starting point. For this, we initialize the population distribution $\mathcal{N}\big(\boldsymbol{\mu}_1 = \boldsymbol{x}_1, \boldsymbol{\Sigma} = \boldsymbol{I}\big)$ and sample $K$ local candidates. Next, we update the mean and covariance of this distribution in a gradient-free way similar to the CMA-ES strategy. We then start a gradient descent from the updated mean $\boldsymbol{s}_1$. The gradient descent runs for at most $J$ steps and is halted if the loss stagnates or increases for more than three steps. The gradient is clipped to avoid leaving the search space boundaries. When the semi-local search reaches a given number of steps (e.g. 100 in our experiments), we update the BO posterior and let BO pick a new starting point $\boldsymbol{x}_2$ globally. We add all the points that semi-local search encounters to the set $\boldsymbol{S}_n$ that is used to compute the posterior for each BO trial.

| **Algorithm 1:** BO with Semi-local Leaps |
| --- |
| **for** $n = 1$ to $max\_steps$ **do**          // Global Search |
|    Compute Gaussian Process (GP) posterior |
|    Get next simulation (control) parameter vector $\boldsymbol{x}_n$ |
|    **for** $i = 1$ to $local\_steps$ **do**     // Semi-local Leaps |
|       Sample $K$ population candidates |
|       Compute descent start $\boldsymbol{s}_1$ from best candidates |
|       **for** $j = 1$ to $J$ **do**     // Gradient Descent |
|          Compute sim. loss and gradients $\nabla_{Sim}\vert_{\boldsymbol{s}_j}$ |
|          Take a gradient step |
|          Break if loss stagnates |

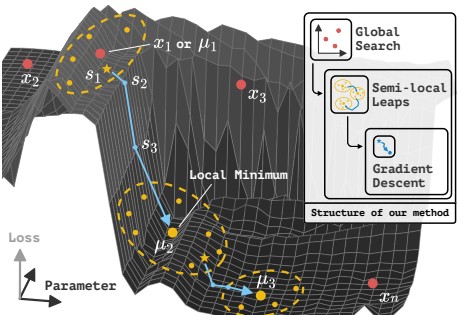

Figure 1: A conceptual illustration of BO-Leap.

Obtaining a noticeable improvement by incorporating gradients into a strategy based on CMA-ES is not trivial. One hybrid approach that seems conceptually sound proposes to shift the mean of each CMA-ES population by taking a step in the direction of the gradient [42]. In our preliminary experiments, taking a single gradient step was insufficient to significantly improve the performance

of CMA-ES: the method from [42] performed worse than gradient-free CMA-ES. In contrast, our semi-local search strategy allows gradient-based descents to take large leaps on the parts of the landscape where gradients are relatively smooth. To prevent being mislead by unstable gradients and avoid wasting computation when stuck on a plateau, our method monitors the quality of the gradient-based evolution of the mean and terminates unpromising descents early.

A number of works combine CMA-ES with randomized restarts, some use BO to select restart points or to initialize CMA-ES, e.g. [43]. Such methods can alleviate the problem of CMA-ES getting stuck in local optima, but do not improve the data efficiency of each CMA-ES run. Using BO for initializing restarts does not make each CMA-ES run more data-efficient. Our insight is to interleave global BO sampling, semi-local population sampling based on CMA-ES equations and gradient-based descents, yielding a tight integration of global, semi-local and local optimization. This allows BO-Leap to tackle loss landscapes that are challenging due to various aspects: local optima, non-smooth losses, and noisy gradients. Experiments in Section 5 show that BO-Leap has strong empirical performance in contact-rich scenarios with highly deformable objects, plastic materials and liquid.

## 4  A Suite of Differentiable Simulation Scenarios

| Framework | Simulation Type | Supported Models | Environments |
|---|---|---|---|
| Nimble [17] | Mesh-based | Rigid | Cartpole, 3-Link Cartpole*, Pinball* |
| Warp [27] | Multiple | Rigid, deformable, fluid | Bounce* |
| DiffTaichi [16] | Multiple | Rigid, deformable, fluid | Fluid*, Swing*, Flip* |
| PlasticineLab [2] | Particle-based | Rigid, deformable | Assembly, Pinch, RollingPin, Rope, Table, Torus, TripleMove, Writer |

Table 1: Summary of differentiable simulation frameworks and environments we use in this work. We used several existing environments and created new ones (marked with asterisks *). See supplementary for details.

In this work, we implement scenarios using several differentiable simulation frameworks. Table 1 summarizes their properties. Our main goal is to study scenarios with deformables, because prior works already explored a number of tasks limited to rigid-body motion (see first paragraph in Section 2). That said, very few works considered contact-rich tasks, for example, even when showcasing differentiable simulation of a highly dexterous hand, work in [44] only shows motion of the hand without touching objects, while the main interest with such hands is their capacity to re-orient objects quickly and precisely while holding the object [45, 46]. One prior work highlighted the potential fundamental limitations of gradients for rigid contacts [6]. Hence, there is a need to further study the fundamentals of how loss landscapes and gradient fields are affected by rigid contacts. For that, we created several scenarios using Nimble [17] and Warp [27] frameworks (described below).

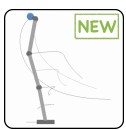

*3-Link Cartpole*: an extension of the classic Cartpole to get more challenging dynamics. Here, a 3-link pole needs to reach the blue target with its tip. Cart velocity and joint torques are optimized for each of the 100 timesteps of the episode, yielding a 400-dimensional optimization problem.

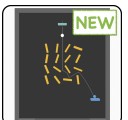

*Pinball and Bounce*: a ball launches and bounces off colliders as in a pinball game. We optimize orientations of $n$ colliders to route the ball from the top to the blue target at the bottom. *Pinball* helps analyze effects of increasing the number of contacts, implemented in Nimble [17]. We also created a simplified *Bounce* scenario to study effects of a single collision in Warp [27], which produced well-behaved gradients in prior work [23].

To go beyond rigid objects, mesh-based simulations can model highly deformable objects, such as cloth. Particle-based simulators can model interactions with granular matter and liquids, plastic deformation with objects permanently elongating, buckling, bending, or twisting under stress. We created several mesh-based and particle-based environments that involve deformables using Diff-Taichi [16] and compared the quality of the loss landscapes and gradient fields they yield. We also included environments from the PlasticineLab [2] that focus on plastic deformation.

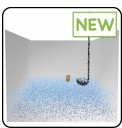 *Fluid*: a particle-based fluid simulation that also involves two rigid objects (a spoon and a sugar cube) interacting in fluid. The objective is to scoop the sugar cube out of the liquid. We optimize forward-back and up-down velocity of the spoon, and let it change five times per episode, yielding a 10-dimensional optimization problem.

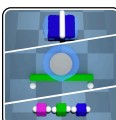 *Assembly, Pinch, RollingPin, Rope, Table, Torus, TripleMove, Writer*: scenarios based on PlasticineLab [2] with particle-based simulations of plastic deformation. We optimize 3D velocities of anchors or pins, allowing them to change five times per episode. This yields a 90D problem for *TripleMove*, 30D for *Assembly*, 15D for the rest.

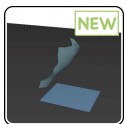 *Swing*: a basic cloth swinging scenario to study dynamic tasks with deformables. We give options to optimize the speed of the anchor that swings the cloth, the cloth width & length, and stiffness of cloth that is partitioned into $n \times m$ patches. Varying $n, m$ lets us experiment with effects of increasing dimensionality of the optimization problem.

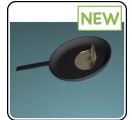 *Flip*: a scenario with highly dynamic motion and rigid-deformable object collisions. The goal is to flip a pancake by moving the pan. We optimize pan motion: $n$ waypoints for left-right, up-down position and pan tilt. The pancake is modeled using a mass-spring model with small stiffness to avoid large forces from dynamic movement.

We built this suite of environments to be representative of a wide range of possible robot manipulation scenarios, regardless of whether differentiable simulators produce sensible gradient in them. We found that some of these environments yield high-quality gradients, while others do not. In Section 5.1, we show that *Cartpole*, *Fluid*, and the eight PlasticineLab environments produce well-behaved gradients and that our method can outperform competing baselines in these scenarios. Then, in Section 5.3, we show that differentiable simulators can produce incorrect gradients in highly dynamic and contact-rich environments like *Pinball*, *Swing*, and *Flip*, which makes gradient-based methods (including our method) less effective in these environments.

## 5 Experiments and Analysis of Optimization Landscapes

In this section, we present visual analysis of optimization landscapes and gradients, as well as comparison experiments, in various environments. For gradient-free baselines, we use Rand (random search), CMA-ES, and two model-free Reinforcement learning methods – PPO [47] and SAC [48]. For gradient-based baselines, we use RandDescents – a method that runs multiple gradient descents with randomly sampled initial points, and BO – using Bayesian optimization to select initial points for gradient descents.

### 5.1 Simulation Experiments and Analysis

We first analyze the usefulness of gradients with rigid objects. The left side of Figure 2 shows experiments on the *1-Link Cartpole* (200 dimensions from cart velocity and joint torques: $(1+1) \times 100$ timesteps). In this case, gradient-free CMA-ES performs similar to gradient-based algorithms (RandDescents, BO), while BO-Leap obtains a significantly lower loss. The right side shows results for *3-Link Cartpole* (400 dimensions: $(1+3) \times 100$ timesteps), which has much more complex dynamics. Here, CMA-ES, PPO, and SAC do better than a completely random search (Rand), but do not bring the pole tip to the target exactly. On the top-right corner of Figure 2, we show qualitative results from a CMA-ES run (top) and from a BO-Leap run (bottom), where BO-Leap brings the tip close to the target.

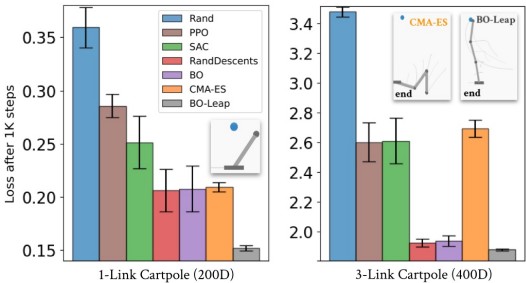

Figure 2: Left: results for *1-Link Cartpole*. Right: results for *3-Link Cartpole*. For all bar charts in this paper (unless otherwise stated): the vertical axis denotes the best loss value after 1,000 optimization steps; the bars show means over 10 runs (for each method), with 90% confidence intervals. Each optimization step runs one simulation episode, computes the loss, and propagates the gradient w.r.t. optimization parameters for algorithms that use gradients.

Next, we study a variety of scenarios that use particle-based simulation. Particle simulators have much larger computational and memory requirements than mesh-based options. Gradient compu-

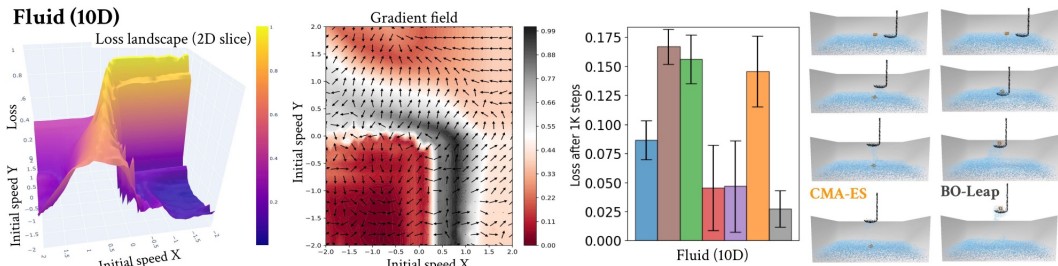

Figure 3: A scenario with scooping up a sugar cube from fluid. The left side shows a 2D slice of the 10D optimization landscape and the corresponding gradients. To make gradient directions visible, we normalize the magnitude of gradients in all gradient plots in this paper; the arrows point towards the direction of negated gradients (i.e. the direction gradient descent updates take). The right plot shows quantitative evaluation of optimization methods. We visualize qualitative results for CMA-ES and BO-Leap on the right side.

tations further increase resource requirements. For the differentiability to be warranted, benefits of gradients have to be significant. We show that gradient-based methods can indeed have large benefits. Figure 3 visualizes the *Fluid* scenario (10 dimensions): the left side shows a 2D slice of the loss landscape. It has many valleys with shallow local optima and appears more difficult than many test landscapes designed to challenge global optimization methods. The middle of Figure 3 shows directions of gradients produced by the differentiable simulator. The right side shows results for gradient-based and gradient free methods. CMA-ES gets stuck in a local optimum: the spoon fails to lift the sugar cube in most runs as shown in the next-to-last column. In contrast, BO-Leap successfully lifts the sugar cube in most runs, outperforming all baselines. BO is designed to balance allocating trials to global exploration while still reserving enough trials to return to the well-explored regions that look promising. In this scenario such global optimization strategy proves to be beneficial. Furthermore, BO-Leap also handles the rough parts of the landscape more effectively than the baseline BO.

In the next set of experiments, we analyze eight environments adapted from the PlasticineLab [2]. Figure 4 shows a *Rope* scenario (30 dimensions), where the objective is to wrap a 'rope' object around a rigid cylindrical pole. The loss landscape is smooth in most dimensions (the left plot shows an example 2D slice). However, higher dimensionality makes the overall problem challenging. The middle plot confirms that gradients are correct in most parts, but also shows a large plateau where even the gradient-based approaches are likely to get stuck. The right side shows evaluation results: CMA-ES and BO fail to wrap the rope fully around the pole, while BO-Leap succeeds in pulling the ends on the back side of the pole correctly.

Figure 5 shows experiments with the other seven PlasticineLab tasks. The top shows the *RollingPin* task (30 dimensions), where the objective is to spread the dark-blue dough using a thin rigid cylindrical white pin (the light-blue region shows the target shape). The loss landscape is smoother than that of the *Rope* task, but has even larger plateaus and flat gradients in most regions, making the problem challenging despite a smooth loss. BO-Leap outperforms CMA-ES, RL baselines and gradient-based methods on this task as well. Example frames on the right show that CMA-ES thins out the dough too much (large gaps appear in the middle, revealing the light-blue target shape underneath). BO and BO-Leap keep the central portion of the dough more uniformly spread.

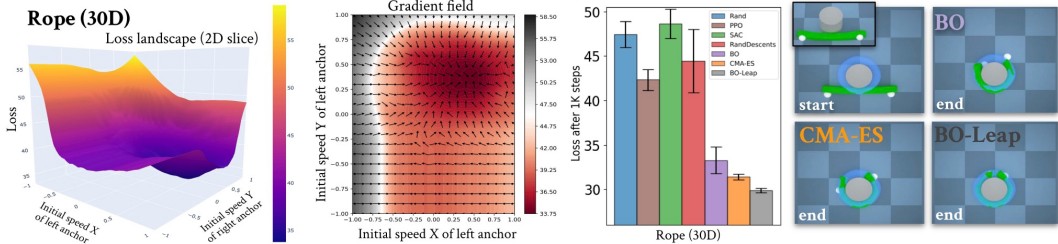

Figure 4: Results for the *Rope* environment. BO-Leap outperforms all other methods and successfully completes the task of wrapping the rope fully around the pole (light blue region shows the target shape). Gradient-free CMA-ES do not discover the optimal behavior, wile BO-Leap can complete the task effectively.

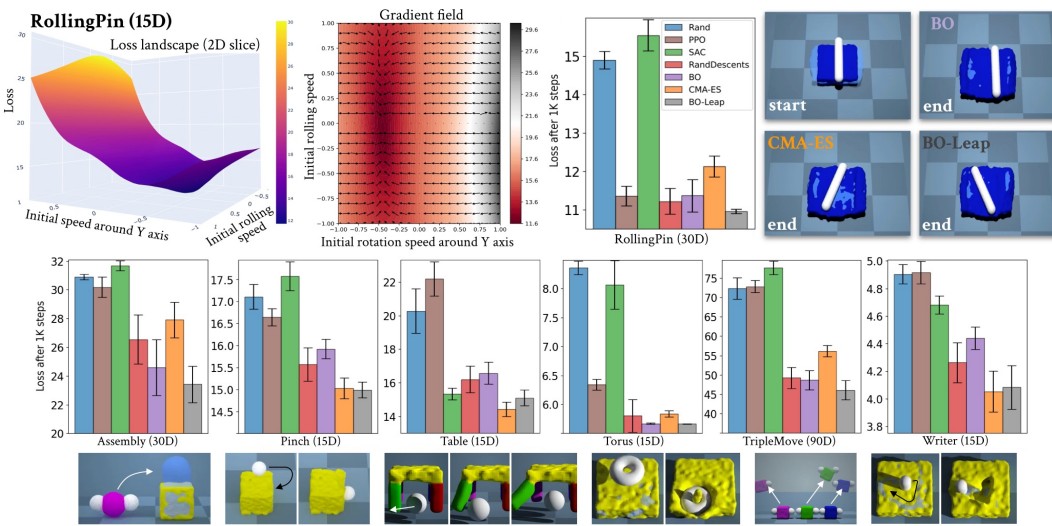

Figure 5: Top row: analysis and results for *RollingPin* task. Bottom rows: results for the other six PlasticineLab tasks. Plots show mean performance over 10 runs of each method per task (see supplemental for more details).

The bottom plots in Figure 5 show results for the additional six PlasticineLab tasks. BO-Leap outperforms gradient-based methods in most tasks, and achieves significantly lower loss than CMA-ES in *Assembly*, *Torus*, and *TripleMove* tasks, while CMA-ES has an advantage in *Table* and a small advantage in *Writer* tasks (see supplementary for further details).

## 5.2 Validation with Real Robot Experiments

For validation on hardware, we first deployed policies found in simulation for the *Fluid* scenario. Figure 6 shows our hardware setup and results. BO-Leap had $> 70\%$ success rate; success rate for RandDescents was $< 60\%$; success rate for was CMA-ES $< 10\%$. This confirms that in reality policies found with BO-Leap can be more effective than those obtained with other methods.

Next, we consider a *real-to-sim* (i.e. parameter estimation) task of identifying the properties of a simulated deformable object to make its motion match the real motion. In this scenario, a Gen3 (7DoF) Kinova robot manipulates a small deformable object by lifting it up from the table surface. The object is tracked by two RealSense D435 depth cameras, one placed overhead, the other on the side. The objective is to optimize the size (width & length), mass, and friction of the deformable object, as well as stiffness of each of the $8 \times 8 = 64$ patches of the object, which yields a 68D optimization problem. The loss penalizes the distance of the simulated corner vertices to the position of the corners marked on the real object. Figure 7 shows results of experiments with two objects: a stiff but flexible paper (left) and a highly flexible cloth (right). In both cases, BO-Leap is able to find physical simulation parameters that produce the best alignment of the simulated and real object. This offers a validation of the method on real data, and shows that it can tackle real-to-sim problems to automatically bring the behavior of simulated objects closer to reality. This is valuable for highly deformable objects, since manual tuning is intractable for high dimensions.

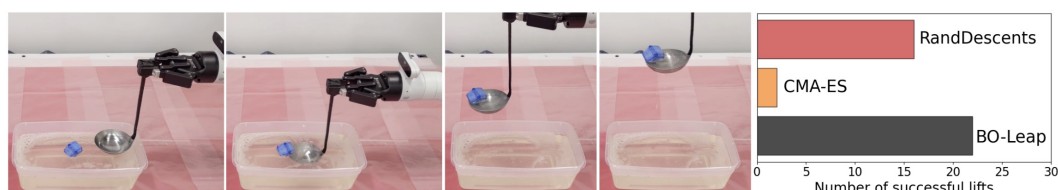

Figure 6: Left: example of a successful lift in *Fluid* scenario on hardware. Right: Evaluation of policies from 30 optimization runs (for each method). We placed the cube in the same start position, and if the cube was lifted out of the liquid – that counted as success. We used foam as 'sugar' for the cube, detergent as 'syrup' and a smaller container for liquid than in simulation. To ensure that this setup mostly matched simulations, we checked that a few lifts that succeeded in simulation still succeeded in reality (and failures still failed in reality).

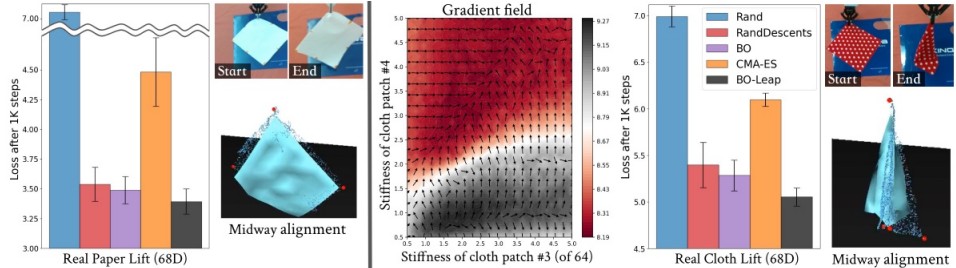

Figure 7: Results for optimizing a simulated object to match the motion of a real paper (left) and cloth (right), which are lifted by a real robot. We visualize the midway alignment showing that our method finds the size, mass, friction and stiffness (for each of the 64 patches) to bring simulated object behavior close to the real one.

## 5.3 Limitations

Modeling global loss posterior with BO can be computationally expensive. We use BoTorch [49] for BO on GPU, which can scale to high dimensions. We focused on results within a budget of 1K optimization steps. If a much larger budget is allowed, then more tests would be needed to validate that BoTorch (or other frameworks) can scale well in terms of compute and memory resources.

The biggest challenges of optimization with differentiable simulators arise due to quality of the gradients, which can be insufficient to be beneficial for gradient-based algorithms, including our method. In Figure 8, we show three environments where gradients produced by differentiable simulators are of poor quality. In the *Pinball* environment, gradients with respect to collider orientations are computable only if a collision (with the pinball) has occurred to begin with. In addition to collision-induced discontinuities, the absence of gradients results in plateaus, affecting gradient-based optimizers. Even in a state-of-the-art simulator Warp [27] with relaxed contact models, a simple *Bounce* task induces gradient discontinuities (see supplemental for this analysis). In *Swing* and *Flip* tasks, while the dynamics appear realistic, differentiable simulators yield gradients with incorrect (often opposite) directions. This is problematic, especially for practitioners who use differentiable simulators without assessing loss landscapes first.

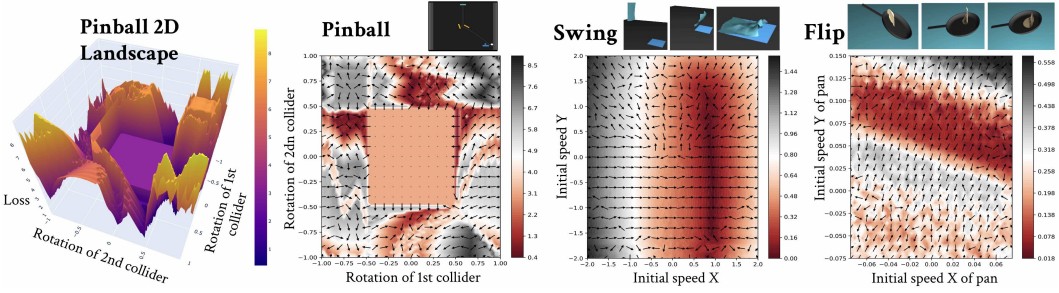

Figure 8: Visual insights into the challenges of obtaining well-behaved gradients for cases with rigid contacts in *Pinball*, and for deformables in the presence of contacts and highly dynamic tasks, such as *Swing* & *Flip*.

## 6 Conclusion and Future Directions

Our analysis shows that differentiable simulation of contact-rich manipulation scenarios results in loss landscapes that are difficult for simple gradient-based optimizers. To overcome this, we proposed a hybrid approach that combines local (gradient-based) optimization with global search, and demonstrated success on rugged loss landscapes, focusing on cases with deformables. We believe our analyses and tools provide critical feedback to differentiable simulator designers and users alike, and would help take differentiable simulators a step closer to real-world robot learning applications.

In future work, it would be interesting to analyze the loss landscapes of simulators learned from data. Model learning methods might not be able to smooth the loss landscapes and gradients if data contains contact-rich interactions. In such cases, optimization algorithms that can make use of gradients and make progress on rough landscapes, such as BO-Leap, would be relevant for the broader problem of global optimization over the dynamics and simulation models learned from data.

**Acknowledgments**

Rika Antonova received support from the National Science Foundation grant No.2030859 to the Computing Research Association for the CIFellows Project. This work was supported in part by a research award from Meta on sample-efficient sequential Bayesian decision making.

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
