# OpenReview forum: "Rethinking Optimization with Differentiable Simulation from a Global Perspective"
_robot-learning.org/CoRL/2022/Conference — CoRL 2022 Oral_

### Official Review · Reviewer_jekJ · 2022-07-13

**Originality:** Excellent
**Technical Quality:** Excellent
**Clarity Of Presentation:** Good
**Impact:** 3

**Recommendation:**

Strong Accept: I recommend accepting the paper and will argue for my recommendation even if other reviewers hold a different opinion.

**Summary:**

Differentiable simulators provide gradients to help system identification and trajectory optimization. Most of the literature on system identification and trajectory optimization uses simple differential simulators (i.e., rigid body), since deformable objects, fluids, strings, and contact-rich environments have non-differentiable areas where gradients are particularly non-smooth.

This paper compares several differentiable simulators (Nimble, Warp, DiffTaichi, Plasticine Lab) that are based on different simulation techniques (mesh-based, particle-based, and a mix of the two) on a large set of challenging environments for both trajectory optimization and system identification.

Furthermore, the authors propose a novel optimization technique for this non-smooth gradient landscape. Such a technique is composed of a global search implemented with a non-parametric Bayesian technique, a semi-local search implemented with CMA-ES, and a local search via gradient descent (RandDescent).

**Issues:**

Missing information
----------------------

My only issue is the lack of a proper definition of the problem:

 - which loss are we optimizing? how is framed the optimization problem?

Writing down a nice problem statement would help the reader understand better the paper.

Avoidable details
-------------------

Some details, instead, are in my opinion unnecessary in the main paper. Algorithm 1 is poorly introduced, and I believe it would require too much space to introduce it properly. Why don't delegate it to an appendix? The explanation in Section 3 is sufficiently clear to me.


I strongly recommend the authors consider these suggestions.

**Quality Of The Limitations Section:**

Limitations are addressed clearly

**Reviewer Expertise:**

3: The reviewer is fairly confident that the evaluation is correct

**Robotics Focus:**

Sufficient demonstration on hardware

**Strengths And Weaknesses:**

Contribution
=========

The contribution is twofold: 1) providing a new analysis of differentiable simulators in contact-rich environments and in presence of fluids/deformable objects and 2) proposing a novel optimization technique that utilizes both gradient and derivative-free techniques.

The contribution, at a high level, is explained very well. The motivation for the work is clear. The placement of the paper in the relevant literature is also clear.

Method
======

At a high level, the method is also clear. However, there are a few important concepts missing.

The authors never formalized or described the trajectory optimization problem and the system identification. This makes the empirical analysis a bit unclear to me. As far as I understood, the trajectory optimization aims to solve a problem similar to the following:

$$
\min_{a_1, ..., a_T} || s_T - g || \quad \\
\text{s.t.} s_0 = \overline{s}_0, \quad \\
s\_{t+1} = f(s_t, a_t)
$$

where $f$ is the simulator. One could use $\nabla_a f(s, a, \omega)$ to solve the problem with gradient descent (or a more complex algorithm like the one described by the authors). I think that will be very beneficial to spend a few lines introducing the problem.

The system identification part is a bit more clear to me, I guess that could be framed as

$$
\min_\omega \mathbb{E}\left[\left(f(s_t, a_t, \omega) - s_{t+1}\right)^2\right]
$$

and $\nabla_\omega f(s, a, \omega)$ can help solve the problem.

In my understanding, this simple ``problem statement'' could be much more useful than the Algorithm~1, which I find honestly confusing.

I agree, instead, with the current high-level description of Bayesian optimization and CMA-ES.

Question
----------

Would the general framework of MPC help to recover bad gradients? I.e., if we take a step in the opposite direction, we could still correct the trajectory in the receding horizon...

Experiments
=========

I am very impressed by the experimental section. It is true that the problem definition that I proposed above would greatly help the reading (at least I think so); but the authors did a great job in introducing the simulators, the environment, and the results.

It is possible to see that the algorithm introduced by the authors exhibits often higher performance (lower cost) in the environment tested. The authors visualize the optimization landscape. I am very impressed by the algorithm being able to solve these problems which are often (relatively) high-dimensional and rich in highly non-linear dynamics.

The real robot experiment (cloth) used for system identification is also very interesting and challenging.

The limitations section is of great value, particularly highlighting the large discontinuities of the gradients of Pinball, Swing, and Flip, preventing a correct optimization. This shows that we still need more accurate gradient estimation for highly discontinuous dynamics. Nevertheless, the potential for robotic application of differential simulators is certainly high.


Details
=====

I think that the term ``policy optimization'' is slightly misused. I'd call policy optimization, methods that improve the parameters of a policy.
The policy, in this case, would be a parametric model that outputs a sequence of actions given a starting state.

I'd call instead ``trajectory optimization'' the process of finding the optimal sequence given a state.

Notice that the main difference is that we can potentially run policy optimization only once, and then re-use our policy for different states, while trajectory optimization needs to be re-run for any new starting state.


The equation in line 89 is very obscure to me: $f(x): f(x^*) = \min_x f(x)$. What does this notation mean??







**Summary Of Recommendation:**

I highly recommend this paper for this conference.

The paper deeply analyzes differentiable simulators and proposes a novel optimization framework to use for both trajectory optimization and system identification.

The analysis is very useful as differential simulators are enjoying increasing popularity in the community due to their potential. Until now, most of the literature was focused on rigid-body dynamics, but this paper analyses challenging scenarios with deformable objects (ropes, cloth), liquid (sugar), and rich contacts between the objects.

I believe that this analysis can be useful for simulator designers, and to understand the current potential and limitations for system identification, trajectory optimization, and policy optimization.

---

> ### Author Response · Authors · 2022-08-23
> **Response to Reviewer4**
>
> Thank you for the review and detailed questions about the problem formulation. We have updated the paper and the supplementary materials based on your feedback. You may find these updated files near the top of the page.
>
> As suggested, we added a paragraph with the problem formulation for policy optimization and real-to-sim to the end of Section 2. This paragraph contains the formal problem definitions, as well as examples that should help the readers understand the formalism.
>
> **For policy optimization:** our formulation supports both reactive and open-loop policies, so the formulation itself is not restricted to trajectory optimization. We can work on adding more examples with reactive policies in the future, if this is an important point to address. Regarding "which loss we are optimizing" - the concrete losses for policy optimization problems are the ones we specified in the supplementary (for each scenario). We hope that this is a bit more clear, after we added "cost (loss)" in several places to remind the readers of this connection.
>
> **For Real-to-Sim:** indeed, this problem can be considered to be identical to system identification, and we adopted your notation for the formal definition. We did not use the term "system identification", because it is commonly used in works that are only concerned with identifying the parameters of the robot. From one of the reviews it became clear that this could cause confusion. In our cloth lifting scenario with a light highly deformable object and a fully actuated robot arm - it does not make sense to estimate the state of the robot. Hence, we only used the term real-to-sim, but please let us know if you think it is crucial to mention relation to system identification in the text.
>
> **Regarding "the equation in line 89 is very obscure":** we expanded the text a bit to make the notation less confusing; we also made sure to connect the notation in the new problem formulation paragraph to the formal description of BO background. We hope that will help to address the comment, but please let us know if we need to clarify this equation further.
>
> We also adopted your suggestion to delegate the full pseudocode version of Algorithm 1 to the supplement and instead included a short outline in the main paper.
>
> **Regarding "would MPC help to recover bad gradients":** investigating how to combine MPC and differentiable simulations that can produce wrong gradients is an interesting direction. However, we did not quite understand what exactly you had in mind specifically for your question. If you would like to discuss this point further, perhaps you could add a clarification?

---

### Official Review · Reviewer_SNTh · 2022-07-13

**Originality:** Good
**Technical Quality:** Very Good
**Clarity Of Presentation:** Very Good
**Impact:** 4

**Recommendation:**

Strong Accept: I recommend accepting the paper and will argue for my recommendation even if other reviewers hold a different opinion.

**Summary:**

Using differentiable simulators for policy optimization and system identification becomes highly non-convex for 'soft' or contact-rich settings.
This work proposes combining gradient-based optimization with black-box search methods to help overcome the challenges of this non-convexity. This approach is evaluted on a diverse range of simulated settings, and validated on a real-world set up.

**Issues:**

Regarding the originality of this idea, I found
Sample and time efficient policy learning with cma-es and bayesian optimisation, LK Le Goff et al. ALife 2020
which is somewhat similar.
I would like the authors to comment on the differences between this approach and theres, and work in a reference into the main text.

**Quality Of The Limitations Section:**

Limitations are addressed clearly

**Reviewer Expertise:**

4: The reviewer is confident but not absolutely certain that the evaluation is correct

**Robotics Focus:**

Sufficient demonstration on hardware

**Strengths And Weaknesses:**

I was generally happy with this submission and it is clear that it is a work of quality. Perhaps there is too much content to comforably sit within the page limit, but that was not a major issue expect that the figure and its labels are quite small to read.

It is no a controversial idea that combining Bayesian optimization and CMA-ES produces a powerful global optimizer. Although I did find some similar, uncited work (see Issues).

Looking at the main text supplementary, I couldn't see results comparing the execution time of the methods considered. It would be good to see these numbers in order to assess performance vs wall-clock time, not just iterations.

Regarding Bayesian optimization (line 88 -- 98), it is does not strictly use non-parametric models, just Bayesian ones. Parametric GPs such as random features approximation and neural networks features have also been used for scalability. Moreover, the 'Var' on line 92 looks like it is in math mode, not text mode.

As a final, somewhat nitpicky comment, I personally dislike 'Rethinking' in a paper title. Most research papers involve rethinking something, so it's a rather unnecessary term. I think a title like 'Global Search in Differentiable Simulatiors with Bayesian Optimization' could be better alternative.

**Summary Of Recommendation:**

I was generally happy with this paper, its contribution, presentation and experiments. 10 seeds is enough for me personally. The codebase also looks good quality. I encourage the authors for a timely release so it can be adopted by the community.

---

> ### Author Response · Authors · 2022-08-23
> **Response to Reviewer3**
>
> Thank you for the review and your questions. We have updated the paper and the supplementary materials based on your feedback. You may find these updated files near the top of the page.
>
> **Regarding "I did find some similar, uncited work":** A number of works combine CMA-ES with randomized restarts, some use BO to select restart points or, as in Le Goff et al, to select points for starting population of CMA-ES. Such methods alleviate the problem of CMA-ES getting stuck in local optima. However, using BO for initializing CMA-ES does not make each CMA-ES run more data-efficient. We do not use full CMA-ES runs, as in work with randomized restarts. Our insight is to interleave global BO sampling, semi-local population sampling based on CMA-ES equations and gradient-based descents, yielding a tight integration of global, semi-local and local optimization. Thank you for suggesting to clarify this point, we added this explanation to Section 3.
>
> **Regarding "results comparing the execution time":**  we did not focus on carefully measuring and comparing the runtime spent in optimization, because for advanced scenarios with deformables, fluids and granular matter the time would be dominated by simulation. It is likely to remain a dominant part, if gradient computation is at least comparable to the forward pass.
> Will differentiable simulators have fast gradients? That part is a bit uncertain, and in this work we did not intend to address this issue. The methods we compared all use a different number of gradient computations, and the speed of these computations is likely to change a lot (and soon) due to rapid advances in differentiable simulation research. For example, the recent version of Warp is supposed to be much faster than even DiffTaichi, and DiffTaichi is already comparatively much faster than many other differentiable simulation frameworks from only a few years ago. But we could not use the recent version of Warp for our experiments, because the current public release still has some key limitations.
> As to the speed of optimization: BO with exact GP posterior will be generally slower than CMA-ES. In contrast, BO with approximate posteriors could be faster than CMA-ES in high dimensional problems. For low-dimensional problems, BO with random feature approximation could be formulated as Bayesian linear regression to run faster than CMA-ES. Hence, there is potential to make BO-Leap runtime competitive, but we didn’t invest any time in this, because running times are dominated by simulation for complex scenarios, especially with particle-based simulations.
>
> **Regarding notation for BO:** as suggested, we removed the word "non-parametric" from the description of BO in related work (we agree that because variational GP approximations, for example, can use NNs to fit the posterior, the overall model could be parametric).
> Thank you for noting the formatting issues with ‘Var’ on line 92, we updated the notation to be \mathbb{V}.
>
> **Regarding "Rethinking" in the title:** this is a response to a large body of recent works on differentiable simulators that focused on scenarios where a single gradient descent was successful. The notion that one just needs to “follow the gradient” seemed to be dominant, so we wanted to draw attention to the fact that the examples in the recent literature could have been skewed. In our opinion, the community needs to rethink which domains, scenarios and tasks should get included as examples of work in differentiable simulation. We wanted to encourage this with our work, in particular by also including examples of simulations where obtaining useful gradients is still a major hurdle.

---

### Official Review · Reviewer_LcC1 · 2022-08-02

**Originality:** Good
**Technical Quality:** Very Good
**Clarity Of Presentation:** Excellent
**Impact:** 4

**Recommendation:**

Weak Accept: I recommend accepting the paper, but will not argue for my recommendation if the majority of other reviewers have a different opinion.

**Summary:**

The paper proposes an algorithm that combines bayesian optimization with evolutionary methods by using local "leaps" in the search space. For gradients, the paper uses differentiable simulators and contact rich and deformable scenarios where the optimization landscape contains many local minima.  The proposed  method does gradent free search as CMA-ES with a population mean and covariance, but uses gradient descent for updating the distribution's mean. The authors apply the method to multiple environments in 4 different differentiable simulators. The environments involve rigid bodies, fluids and deformable objects. The results show that BO-leap performs better than both CMA-ES and Bayesian Optimization and the authors illustrate the gradient field and why other algorithms do not perform well. Last, the authors apply the method to a real robot for a manipulation of a deformable cloth, but the method is used for finding "simulation parameters" to match the real robot's behavior.

**Issues:**

Hardware experiments can be improved (please see above).

**Quality Of The Limitations Section:**

Limitations are addressed clearly

**Reviewer Expertise:**

4: The reviewer is confident but not absolutely certain that the evaluation is correct

**Robotics Focus:**

Sufficient demonstration on hardware

**Strengths And Weaknesses:**

The paper proposes a method that combines strengths of 2 different methods. The authors do a great job at explaning the weaknesses of both methods with examples gradient fields for existing problems. I like the diversity of the tasks, environments and simulators that the authors test their algorithm on. The examples very clearly show when the proposed algorithm would succeed while others fail. The illustrations of the results are very well presented and described.

I'm slightly suprised by the results on Figure 2 (left side). I would expect CMA-ES to achieve better results with the right hyperparameters or initialization.

The application to the real hardware is actually to optimize simulation parameters to match the real robot behavior. So it is not really optimizing the behavior as it's done in simulation. We also cannot call it optimization of the simulator as it optimizes for only one real world example, so it's likely to fail at different poses. Nevertheless it shows one way to use the algorithm with a real robot, so I think it's ok. I would love to see the algorithm used to optimize a policy that is deployed to the robot. Also, I'm very curious about the potential gains on the learning curve (i.e. number of evaluations) for these tasks.

**Summary Of Recommendation:**

The paper proposes a new algorithm (to the best of my knowledge) and showcase many different type of problems where the algorithm performs better than vanilla CMA-ES or BO. The paper is overall great and the simulation results are also great, but I think that the hardware evaluation could be slightly different (i.e. optimization of a policy) or if it is tuning of a simulation, it could be based on hundreds of real robot experiments and see if the optimized simulator matches the behavior in a holdout set. Otherwise, the paper is very nice in my opinion.

---

> ### Author Response · Authors · 2022-08-23
> **Response to Reviewer2 - Part 1 of 2**
>
> Thank you for the review and feedback. We have updated the paper and the supplementary materials based on your feedback. You may find these updated files near the top of the page.
>
> Below, we address your concerns in detail:
>
> **"Would love to see the algorithm used to optimize a policy that is deployed to the robot":** Based on your and Reviewer1’s suggestions, we deployed the policy learned by BO-Leap and several baselines on hardware for the Fluid scenario. We updated Section 5.2 to describe the evaluation of the policies from 30 optimization runs. These experiments corresponded to training in simulation with 30 different seeds for each algorithm. Since we wanted to provide this evaluation as soon as possible, we did not set up an extensive vision system to track the motion of the cube. We placed the cube in the same position at the start of each evaluation run, and if the cube was lifted out of the liquid - we counted that as success. BO-Leap had 22/30 successful runs (73% success rate); RandDescents : 16/30 (53%); CMA-ES : 2/30 (7%). This demonstrates that the policies obtained by BO-Leap can be effective on hardware. We also uploaded a short video comparing several runs of BO-Leap and CMA-ES (please see the attachment).
>
> We hope that the above serves as a good example of the capability of our approach to find policies that succeed in reality. We would also like to clarify that “real-to-sim” (which we addressed with our initial hardware experiments) is a separate problem, and it is also important for cases with highly deformable objects. The goal of real-to-sim, as you mentioned, is “finding simulation parameters”. In manipulation, the focus is often on making the motion of the simulated objects match reality (not the motion of the robot as is often the focus for locomotion). For a fully actuated rigid manipulator arm with accurate controllers it could be easy to align simulation of the robot with reality. Highly underactuated robots and soft robots would present a challenge for real-to-sim. We focussed our hardware experiments on the behavior of a highly deformable object (not the robot behavior). Please let us know if this was not clear in our description of the hardware experiments, and we would work on clarifying that in the text of the submission.
>
> **"We… cannot call it optimization of the simulator as it optimizes for only one real world example, so it's likely to fail at different poses", "if it is tuning of a simulation, it could be based on hundreds of real robot experiments":** We apologize if our description of the real-to-sim problem was not sufficiently clear and our choice of evaluation caused the confusion. The goal is indeed to optimize the simulator by finding the properties of the cloth (cloth mass, friction properties, bending properties for each cloth region). The reason we aimed to infer these from a single trajectory was to demonstrate that there is no need for expensive data collection on hardware, and that with differentiable simulators such properties can be inferred almost in real time from just one short trajectory. With that, we could transition to policy optimization in simulation of any task that involves a deformable object whose properties we have inferred during real-to-sim. Policy optimization is then a separate problem, since it can involve almost any tasks with the deformable object whose properties we just inferred. The limitation is that we might not be able to infer all object properties from a single trajectory, e.g. if the robot does not stretch the cloth, we would not infer cloth stretching properties. This problem of the need to identify all motion types needed to infer all object properties is a general problem for real-to-sim.
>
> **"I would expect CMA-ES to achieve better results with the right hyperparameters or initialization":** CMA-ES does not use gradients, so it is reasonable to expect gradient-based approaches to perform better for cases with smooth motion, where it is easy to obtain high-quality gradients. CMA-ES also does not scale as well with the increased dimensionality of the problem. In Figure 2, CMA-ES does on par with the gradient-based methods for a 100D problem, but does significantly worse on the 400D problem. It could be possible to conduct additional hyperparameter tuning for CMA-ES. However, it is not guaranteed that hyperparameters that work in one part of the landscape (e.g. rugged) would be optimal in other parts (e.g. large platoes). Hence, there might not be a single optimal set of hyperparameters across the search space even for one given environment. Furthermore, it is not certain that CMA-ES would still remain data-efficient if we count the samples needed to optimize hyperparameters as part of the “budget”.
>
> **"I'm very curious about the potential gains on the learning curve":**
>
> _Please see our response in the next (sub) comment ->_

---

> > ### Author Response · Authors · 2022-08-23
> > **Response to Reviewer2 - Part 2 of 2**
> >
> > **Comment:**
> >
> > **"I'm very curious about the potential gains on the learning curve":** Please see the image with 6 plots attached to this response (in .zip archive). We plotted learning curves for scenarios from Figures 2,3,4. From Figure 5 we plotted RollingPin (visualized in the paper) and two cases where BO-Leap was worse than CMA-ES at 1K trials - Table and Writer. To make this organized, we plotted one method from each category : SAC (from RL methods), RandDescents (gradient-based baseline), CMA-ES (gradient-free baseline), BO-Leap (our method). PPO and SAC had a comparable (bad) performance, but SAC got close to the best-performing method in the Table scenario, so we included SAC. Baseline BO was similar to RandDescents in most cases - it only did significantly better than RandDescends in Rope scenario, but much worse than BO-Leap, so we omitted baseline BO to avoid clutter. Overall, the plots show that on smooth loss landscapes (e.g. 3-Link Cartpole, RollingPin) gradient-based methods get an almost immediate advantage. For rugged landscapes, such as Fluid, the gains are more gradual. BO-Leap has a very large lead on Rope, where the landscape is smooth, and RandDescents gets completely stuck on plateaus. In the two cases where BO-Leap is not the overall winner, i.e. the Table and Writer scenarios: BO-Leap has strong initial gains, but CMA-ES gets a slightly better loss eventually.
> >
> > **Zip File:**
> >
> > /attachment/ee76bfddf2840be592bdbf860a19c49f3b6537f1.zip

---

> > > ### Author Response · Authors · 2022-08-23
> > > **Video illustrating new hardware experiments**
> > >
> > > **Comment:**
> > >
> > > This is a short video illustrating the new hardware experiments that we described in the response.
> > >
> > > **Zip File:**
> > >
> > > /attachment/64330a6870f78cc5fd5a904736cb87b391a50d4b.zip

---

### Official Review · Reviewer_LRvk · 2022-08-08

**Originality:** Very Good
**Technical Quality:** Very Good
**Clarity Of Presentation:** Very Good
**Impact:** 3

**Recommendation:**

Strong Accept: I recommend accepting the paper and will argue for my recommendation even if other reviewers hold a different opinion.

**Summary:**

In this work, the authors focus on the challenges of using differentiable simulation for learning optimal policies in tasks involving nonlinear dynamics, such as in contact-rich tasks and those involving deformable objects. The authors introduce a suite of benchmarking tasks to analyze the performance of various state-of-the-art differentiable physics simulators and gradient-based policy optimization methods. Through their study, the authors find that, often in such scenarios, the optimization landscape contains many discontinuities and spurious local optima, causing the gradient-based methods to struggle in finding globally optimal policies. To address this, the authors propose a novel Bayesian optimization scheme that uses semi-local ‘leaps’ to obtain a global search method that can use gradients effectively while being robust to noisy gradients. The authors evaluated the proposed scheme in a suite of experiments, where it outperformed the baseline methods.

**Issues:**

- Rephrase the third paragraph (Lines 36-47) to make the contributions of the work clearer (same as listed in the video). The sections of the paper (sections 3 and 4) can also be rearranged to follow this list, which should enhance the overall readability of the manuscript.
Sentences in Lines 56-59, starting with “Most modern simulators…” require rephrasing for clarity.
- Add some exemplary citations of each of the families of global optimization methods (Lines 66-70).
- Cite the relevant work to support the claim about CMA-ES: “This method is technically not fully global — while it can overcome 86 shallow local optima, it can get stuck in deeper local optima”.
- In section 4 para 1, please cite which previous section is being referred to in Line 148. Also, please cite the relevant work that considers contact-rich tasks in the following line (Line 149).
- While describing each of the experiments in section 4, consider explaining the dimensionality of the optimization problem in terms of the DOF considered for the task. For example, for the 3-link cartpole task, say that the optimization problem is 400-dimensional (cart velocity and joint torques (1+3) * 100 timesteps).
- In section 5, please include in the text that CMA-ES performs better than BO-Leap in the table and writer tasks and provide a brief discussion on why that may be the case.
- Including a real-world task of a robot manipulating deformable objects using the policy learned with BO-Leap and comparing it with other state-of-the-art methods ( PPO, SAC, [1], [2]) would demonstrate the effectiveness of the proposed policy optimization scheme better.


References:
- [1] Hang et al.. "Modeling, learning, perception, and control methods for deformable object manipulation." Science Robotics 2021.
- [2] Zeng, et al. "Transporter networks: Rearranging the visual world for robotic manipulation."  CoRL 2020


**Quality Of The Limitations Section:**

Limitations are addressed clearly

**Reviewer Expertise:**

3: The reviewer is fairly confident that the evaluation is correct

**Robotics Focus:**

Highly relevant to robotics but no hardware experiments

**Strengths And Weaknesses:**

Strengths:
- The authors study challenges in using the differentiable simulation to find optimal policies for complex tasks and identify plausible reasons behind them. Understanding these bottlenecks will help the community to build better physics simulators and policy learning algorithms that can help the robots to perform complex manipulation tasks in the future.
- The figures included in the manuscript are rather well-made and illustrate the ideas discussed comprehensively. Figure 1, illustrating the BO-Leap algorithm, describes the algorithm in a succint yet effective fashion to the readers. Figures visualizing optimization landscapes (Fig. 3, 4, 5, and 7) help the readers to better appreciate the challenges faced by the policy optimization methods in such tasks.
- I liked that the authors also briefly described each step of Algorithm 1.
- Table 1 summarizing differentiable simulation frameworks, their type, supported models, and environments is quite useful to the readers to understand the capabilities of the available state-of-the-art physics simulators.

Weakness:
- A comparison of the proposed method, BO-Leap, with other standard policy optimization methods such as PPO, SAC, etc., should also be included.
- The current manuscript requires some reorganization and updates for better readability.
- Real-world tasks of a robot manipulating deformable objects using the policy learned with BO-Leap would demonstrate the effectiveness of the proposed policy optimization scheme better.


**Summary Of Recommendation:**

The authors study an important problem of understanding the challenges in using the differentiable simulation to find optimal policies for complex tasks. The manuscript presents the core ideas successfully but requires some further work. The authors must address a few issues before I can feel confident in recommending the manuscript to be accepted at the conference. If the authors address the raised concerns in their rebuttal satisfactorily, I would be happy to recommend the manuscript for publication.

---

> ### Author Response · Authors · 2022-08-23
> **Response to Reviewer1 - Part 1 of 2**
>
> Thank you for the review and the detailed list of issues. We have updated the paper and the supplementary materials based on your feedback. You may find these updated files near the top of the page.
>
> Below, we first address the three main weaknesses:
>
> **a) Additional Baselines:** We ran PPO and SAC baselines on all the scenarios for policy optimization and included the performance of these algorithms in the bar plots in the main paper. Please see the updated Figures 2, 3 and 4. We parameterize the policies in the RL baselines by 3-layer MLPs with hidden layer size 64, and train them so that they experience the same amount of environment interactions as other baselines. Additional implementation details can be found in Section C of the supplementary.
>
> **b) Re-organization and Citations:** We addressed comments regarding re-organization and citations in the manuscript. Please see the changes highlighted blue in the manuscript (with notes in the margins that start with “Note to Reviewer1”).
>
> **c) Hardware Experiments:** As suggested, we deployed the policy learned by BO-Leap and several baselines on hardware for the Fluid scenario. We updated Section 5.2 to describe the evaluation of the policies from 30 optimization runs. These experiments corresponded to training in simulation with 30 different seeds for each algorithm. Since we wanted to provide this evaluation as soon as possible, we did not set up an extensive vision system to track the motion of the cube. We placed the cube in the same position at the start of each evaluation run, and if the cube was lifted out of the liquid - we counted that as success. BO-Leap had 22/30 successful runs (73% success rate); RandDescents : 16/30 (53%); CMA-ES : 2/30 (7%). This demonstrates that the policies obtained by BO-Leap can be effective on hardware. We also uploaded a short video comparing several runs of BO-Leap and CMA-ES (please see the attachment).
>
> We hope that the above addresses the main weaknesses. Below we respond to the list of more detailed issues provided by the reviewer:
>
> **1)** As suggested, we re-organized the ending of the introduction to follow the order of contributions as in the video and included a brief outline for clarity.
> Regarding the order of Section 3 (algorithm) and Section 4 (environments) - our idea was to have the continuity between the Background section and Section 3 (algorithm), since the BO algorithm background would then be easy to refer to when reading Section 3, and would also be fresh in memory for those who read the text in order. Similarly, we thought that it would be good for Section 4 (environments) to be right before Section 5 (Experiments and Landscapes), because this way the details about environments will be fresh in memory for those who read the text in order.
> As suggested, we clarified the text that used to be in lines 56-59.
>
> **2)** We added citations for examples of algorithms from each of the families of global optimization methods.
>
> **3)** We added a citation for “A comparative study of CMA-ES on large scale global optimisation” by Omidvar et al. The most relevant bits from the conclusion are: “CMA-ES is designed as a highly competitive and robust local optimiser. [..] CMA-ES suffers from the curse of dimensionality [..] This is clearly evident from the performance of CMA-ES on multimodal test functions.”
>
> **4)** We clarified the section reference in the text. We also added a note about the lack of contact-rich tasks in literature on differentiable simulation (and cited examples of contact-rich tasks in robotics).
>
> **5)** We included the update to environment descriptions in Section 4 as suggested (highlighted in blue in the updated pdf).
>
> **6)** We added the text noting CMA-ES performance on the Table and Writer tasks. By just looking at the slices of the loss landscapes we plotted so far and inspecting visualizations of the environments, it is not immediately clear why CMA-ES is slightly better in the Writer and Table tasks. We want to avoid hypotheses that would not be clearly supported by the data, hence, for now, we just stated the fact about CMA-ES’ performance on these tasks in the text, without making further comments. We could conduct a more in-depth analysis of the loss landscapes and the optimization dynamics of CMA-ES specifically on these tasks, if needed.
>
> _Notes about comparison with PPO, SAC, [1], [2] are in the next (sub) comment ->_

---

> > ### Author Response · Authors · 2022-08-23
> > **Response to Reviewer1 - Part 2 of 2**
> >
> > **7)** Regarding RL baselines: please see our response in a) above, and the Figures 2, 3 and 4 in the main version of the paper that now include comparisons with PPO and SAC. The performance of RL was generally worse than that of gradient-based methods (here, by gradient-based we mean the ones that leverage the differentiability of the simulators). This is not particularly surprising, since model-free RL does not aim to prioritize data efficiency. It has been shown that on some control problems that RL can be easily outperformed by e.g. model-based methods in a low-data regime [Recht2019].
> >
> > Regarding comparison with [1] Hang et al.. "Modeling, learning, perception, and control methods for deformable object manipulation":, we could not easily identify the most relevant method in this broad survey. Most methods mentioned there were evaluated on specific tasks, not a broad set of tasks with a variety of materials. So, we decided to prioritize the comparison with PPO and SAC, as suggested, since these are the most widely used and versatile RL methods.
> >
> > As to [2] Zeng, et al. "Transporter networks: Rearranging the visual world for robotic manipulation" : Section 3.3 of [2] states that they “assume access to a dataset of n expert demonstrations”. This is quite different from the setup we are aiming for. We would like to train without any access to demonstrations. Instead, we propose to leverage simulation to provide enough structure for effective and autonomous discovery of optimal policies. For future work we could look into combining our method with parts of the architecture from [2] for handling RGB inputs, but in our view that would be a different research focus.
> >
> > _[Recht2019] B. Recht. “A Tour of Reinforcement Learning: The View from Continuous Control”. Annual Review of Control, Robotics, and Autonomous Systems. 2019_
> >
> >
> > Overall, we would also like to ask the reviewer for additional advice. This review seemed to be focused on the aspect of our work that addresses feedback for policy learning. However, we intended to also address the real-to-sim problem as it eliminates the need for manual simulation parameter tuning. This is important for scenarios with highly deformable objects, fluids and granular matter where tuning by hand is frequently intractable. This is also why we prioritized including a real-to-sim hardware experiment in the original submission. Does the reviewer have any suggestions on how to improve our paper such that the explanation of the real-to-sim problem and its relevance is more clear?

---

> > > ### Author Response · Authors · 2022-08-23
> > > **Video illustrating new hardware experiments**
> > >
> > > **Comment:**
> > >
> > > This is a short video illustrating the new hardware experiments that we described in the response.
> > >
> > > **Zip File:**
> > >
> > > /attachment/cdb6acdf83ea160dc17e8a4af73d7785458d591b.zip

---

### Author Response · Authors · 2022-08-23
**Updated submission pdf and supplementary materials**

**Comment:**

Based on the feedback from the reviewers, we have updated the pdf of the submission (icon above), the supplementary and the video (zip file below). We thank the reviewers for their constructive feedback.

**Zip File:**

/attachment/d5aa108b789065de324acba449e24252354639d5.zip

---

### Meta-Review · Area_Chair_Kceq · 2022-08-13

**Recommendation:** Accept (Oral)
**Confidence:** 4

**Metareview:**

The paper proposes an optimization method that combines Bayesian optimization as the global search method and gradient based method using differentiable simulation. The method is demonstrated on an extensive set of experiments including rich contact and deformables, and is also validated on a real robot.

Most of the reviews agree that

(1) The work has excellent presentation quality, and does a good job at motivating the work.

(2) The experimentation is impressive with diverse tasks, environments, and simulators.

(3) The novelty of the optimization framework is good.

They also brought up some missing citations, and the hardware experiment improvement suggestions.

During the reviewer discussion, there is a general acknowledgment of the good quality of the paper among most reviewers. Therefore, we recommend accepting the paper as oral.

---

> ### Author Response · Authors · 2022-08-27
> **Response to Meta Review**
>
> Thank you for the meta review. At the beginning of the week, we responded to each reviewer and did our best to address all of their questions. The main updates were:
> - We completed the hardware experiments that Reviewer 1 and 2 wanted to see. We updated the paper PDF with results and provided the video of experiments to the viewers.
> - We also added RL baselines that Reviewer 1 requested.
> - We included the citations that reviewers brought up and discussed their relation to our work.
>
> We hope that we were able to address all the questions fully. We have not received any further notifications, but in case we missed anything please let us know.